# Adolescent Girls' Experiences Regarding Teenage Pregnancy in the Rural Villages of Limpopo Province, South Africa

**Patrone Rebecca Risenga * and Sheillah Hlamalani Mboweni**

Departmental of Health Studies, University of South Africa, Pretoria 0002, South Africa
* Correspondence: risenpr@unisa.ac.za

**Abstract:** Every year, 7.3 million girls become pregnant before they turn 18. Teenage pregnancy increases when girls are denied the right to make decisions about their sexual health and well-being, which is a gender equality issue. Among the challenges of gender equality are those expectations that communities have about girls and early motherhood, sexual violence, and rape. Another challenge is the early marriages of children to older men coupled with the unique risks faced by these girls during pregnancy, for example, the interruption of their education, health risks, such as HIV, premature birth, and increased maternal mortality, denying the girls the right to live a healthy life. This study sought to explore the experiences of adolescent girls regarding teenage pregnancy in the rural villages of the Mopani District, Limpopo. A descriptive, explorative, and qualitative design was followed to collect data from 20 pregnant teenagers in a 13–19 years-old age group. A nonprobability purposive sampling method was used to select the participants from the three villages of the Mopani District. The data were collected using an in-depth individual interview. Tesch's eight steps of data analysis were also applied. The study findings reveal several factors that explain the high rates of teenage pregnancy in rural Limpopo. Among these are the socioeconomic and cultural factors that predispose teens to pregnancy. The consequences of teenage pregnancy were expressed in terms of regret and ill health.

**Keywords:** teenage pregnancy; gender inequality; adolescent girls

## 1. Introduction and Background

Every year, 21 million girls aged 15–19 years in low- and middle-income countries (LMICs) become pregnant, and of these, approximately 12 million give birth. A total of 777,000 births occur among adolescent girls younger than 15 years [1]. Teenage pregnancy increases when girls are denied the right to make decisions about their sexual health and well-being, which is a gender equality issue [2]. Teenage pregnancy is a public health concern in high-income countries (HICs) and LMICs and has social consequences for unmarried pregnant teenagers, which include stigma, rejection, or violence by partners, parents, and peers. In addition, girls who become pregnant before the age of 18 are more likely to experience violence within a marriage or partnership [1]. Furthermore, complications during pregnancy and childbirth are the leading cause of death of 15–19-year-old girls globally [1]. Of the approximately 5.6 million abortions that are carried out each year among adolescent girls aged 15–19 years, 3.9 million are unsafe, contributing to maternal mortality, morbidity, and lasting health problems. The World Health Organization also states that adolescent mothers (aged 10–19) face higher risks of eclampsia, puerperal endometritis, and systemic infections than women aged 20 to 24, and the babies of adolescent mothers face higher risks of low birth weight, preterm delivery, and severe neonatal conditions [1].

South Africa has a large proportion of young people, and teenage pregnancy has emerged as one of the major public health problems. A shocking 2020 report from Statistics South Africa shows that there were 33,899 adolescent mothers, representing 17.1% of the

population, and 660 babies birthed by girls aged 10 years or under [3]. The teenage pregnancy statistics of girls between 10 and 17 years reveal that Kwa-Zulu-Natal is the leading province for teenage pregnancy, followed by Limpopo in South Africa [3]. Barron et al. [4] conducted a retrospective study using data from three districts on health information from 2017–2021 and discovered that teenage pregnancy among 10–14-year-old girls increased by 48%, and the birth rate per 100 girls in this age category increased by 1.1 to 1.5 [4]. The study also mentions that teenage pregnancy among adolescent girls aged 15–19 increased by 17.9%. These rates are higher in rural areas such as Limpopo, Mpumalanga, and the Eastern Cape, as well as in urban areas such as Gauteng and the Western Cape.

According to Lindroth [5], young women in low-income countries were commonly forced to go through with unplanned pregnancies and suffer from sexually transmitted infections, which might result in them falling victim to gender-based violence [5]. The study also highlights that the majority of the world's adolescents lack enough knowledge of sexual reproductive health rights, which makes them more exposed to unplanned pregnancies, sexually transmitted infections, and sexual violence [5]. Another study conducted by Moult & Müller [6] further reveals that sexual and reproductive health (SRH) is far from settled by either the healthcare professionals who provide these services or by young people who must seek them [6]. Without sexual and reproductive health knowledge, teenagers are more vulnerable to teenage pregnancy and its complications.

Chapter 2 (section 9(3)) of the Constitution of the Republic of South Africa of 1996 describes that the state may not unfairly discriminate directly or indirectly against anyone on one or more grounds, including race, gender, sex, pregnancy, marital status, ethnic or social origin, sexual orientation, age disability, religion, conscience, belief, culture, or birth, which includes pregnant learners. However, this does not mean that the state should keep quiet while the future of a child girl is affected. Measures must be put in place to deal with pregnancy as an issue affecting teenage girls. A study conducted by Gibbons et al. [7] indicates that, regardless of advances in access to education at primary and secondary levels, women still face significant challenges, such as their ability to engage in unpaid work and to be employed in the informal sector [7]. This also makes it more likely that South African women will live in poverty and become victims of interpersonal violence. Therefore, this study sought to investigate adolescent girls' experiences regarding teenage pregnancy in the rural villages of Limpopo. The key research question is, what are adolescent girls' experiences regarding teenage pregnancy in the Mopani District?

*1.1. Problem Statement*

Between April 2017 and March 2018, a total of 16,238 children were born to adolescents in the province of Limpopo's state-owned hospitals. Teenage pregnancy rates escalate daily, and Limpopo has the second-highest rate in South Africa. Limpopo reported 5954 teenage pregnancies in 2020, birthed by teenagers between 10 and 17 years old [3]. There is an outcry against teenage pregnancy in the Mopani District, where even primary school teens are falling pregnant. In July 2017, one primary school in Mopani had two teenagers in the age group 14 to 16 years who were pregnant [8], and in one high school in the same vicinity, 37 teenagers were pregnant [8]. In July 2018, the researcher visited a primary school where four Grade 5 learners, who were younger than 14, were mothers. This reflects the crisis faced by the Mopani District regarding teenage pregnancy. School teachers complain about absenteeism, specifically among those girls who are reported to be pregnant, and absenteeism leads to poor academic achievement. If the rate of teenage pregnancy keeps on rising, what will the implications be for the sexual and reproductive health rights of women in the Mopani District in the future? Will teenage pregnancy cripple the educational and economic aspects of future families by harboring poverty? This has a bearing on the educational and economic level of women in South Africa. Something must be done to address the challenges related to teenage pregnancy among girls in the Mopani District, hence the need for this study.

*1.2. Purpose of the Study*

The study was conducted to explore the experiences of adolescent girls regarding teenage pregnancy in the rural villages of the Mopani District, Limpopo.

## 2. Materials and Methods

The research design was qualitative, descriptive, and explorative in nature.

*2.1. Study Setting*

The study was conducted in the clinics of four different villages in the Mopani District of Limpopo. Mopani District Municipality comprises five local municipalities: Greater Letaba, Greater Giyani, Greater Tzaneen, Maruleng, and Ba-Phalaborwa. The local municipalities are further demarcated in terms of wards, totaling 118 wards in the entire district, with 15 urban areas (towns and townships) and 348 villages (rural settlements). The study was conducted in Greater Giyani Municipality villages. In 2016, Limpopo had a population of 5,726,800, which is 10.4% of the South African population.

*2.2. Population*

The target population in this study consisted of all the pregnant teenagers in the selected rural villages of the Mopani District. Inclusion criteria: female adolescents between the ages of 13 and 19 years who were residing in the selected municipality and villages.

*2.3. Sampling Method*

A nonprobability purposive sampling method was used to select the study participants. The method was relevant because only pregnant teenagers were able to provide their lived experiences.

*2.4. Sample Size*

A total of 20 pregnant teenagers in the age group 13 to 19 years were interviewed. The sample size depended on data saturation. The point of data saturation was reached after 20 interviews, and it was no longer necessary to continue with the interviews.

Creswell [9] refers to saturation as getting similar data or ideas repeatedly during interviews in a qualitative study, which signifies the completion of data collection for a particular phenomenon [9].

*2.5. Data Collection Process*

The 20 interviews were conducted at a clinic. An empty private room at the clinic was used for data collection. The researcher introduced the question, and an audiotape was used with the participants' permission. Each interview lasted 45 to 60 minutes. All the participants were Tsonga and were interviewed in Xitsonga. They were from three different villages in Mopani District, Limpopo, South Africa. The responses were paraphrased and summarized, and the researcher probed for answers for data collection.

Measures to Ensure Trustworthiness

Measures to ensure the trustworthiness of the study were credibility, transferability, dependability, and confirmability. Credibility was ensured by having prolonged conversations with the participants as well as recording their responses. Transferability was ensured through a detailed description of the background information of participants and the setting for the study. The researcher ensured dependability by describing the methods and comparing field notes with voice recordings. Verification of data against the recorded responses was done in drawing the study findings. Confirmability was ensured by involving a different researcher during data collection and data analysis to verify interpretations, data verification recommendations, and conclusions. Clear descriptions of each stage of the research process, explaining and justifying what was carried out by the voice recording, and

the verbal quotes recorded during the unstructured individual interviews, are all evidence to ensure authenticity [9].

*2.6. Ethical Considerations*

Ethical approval for the study was received from the Ethics Committee of the University of South Africa (Ethical clearance number: Reference #: 90187598_CREC_CHS_2021). Ethical considerations were adhered to in order to protect the participants, as is presented in the subsequent descriptions. Permission was obtained from Greater Giyani Municipality in the Mopani District, Limpopo, to conduct the study.

The participants who were above 18 years signed the consent form after a thorough explanation of the study. Permission was obtained from the parents of those participants who were under the age of 18, and the signing of assent was facilitated. Voluntary withdrawal from the study at any point was emphasized. Confidentiality, anonymity, and privacy were maintained by using codes instead of the names of the participants. The researcher continuously played back the tapes to the participants to validate what had been recorded. The transcripts of two of the participants were later read to confirm the interviews.

The researcher suspended information known to her about teenage pregnancy to avoid misinterpreting the phenomenon experienced by the participants by using bracketing. Bracketing is a scientific procedure for when a researcher puts their presuppositions, prejudices, assumptions, hypotheses, or prior experiences on wait in order to observe and describe the phenomenon [9]. Bracketing made it possible for the researcher to focus on the participants' views and to shape the data collection process according to them. The reason for bracketing was to reduce a biased view and any preconceived ideas the researcher could have had.

*2.7. Data Analysis*

In this study, Tesch's eight steps were used for the data analysis [9]. Themes, categories, and subcategories were developed according to the data obtained from the individual interviews. The steps taken during data analysis were as follows: the first step was to make sense of everything. This involved reading through all the transcriptions carefully and jotting down ideas as they came to mind. The researcher studied all the transcriptions attentively to make sense of them. She picked one document (interview) and went through it to check its meaning. Topics were formed from the data obtained, and clusters of similar topics were formulated. The formulated tasks were abbreviated as codes and written next to the appropriate segments of the text. Verification was carried out to check whether new categories and codes had emerged. Categories were formed from each theme that had been developed and presented in the findings. A final decision on the abbreviation for each category was made, and codes were written alphabetically. Data material belonging to one category was assembled in one place, and a preliminary analysis was carried out. Lastly, the existing data were recorded [9].

**3. Results**

Four themes, each with respective subthemes, emerged: (1) sexual abuse and other causes of teenage pregnancy; (2) socioeconomic factors–a license for teenage pregnancy and freedom; (3) cultural factors, to which adolescent girls are susceptible, and (4) the consequences of teenage pregnancy.

*3.1. Theme 1: Sexual Abuse as Other Causes for Teenage Pregnancy*

This theme did not have a subtheme. The participants indicated that they were sexually abused and raped by their fathers and stepfathers. Another participant revealed that she was forced to have sex with the traditional healer.

> *"I got pregnant after having sex with my father because he believes his ancestors told him that before all his daughters get married, they must have sex with him to teach us what to*

*expect out there. You won't believe that our mother knows about it, and she keeps quiet, so nothing was done."*

*"I was locked in the same room with my stepfather, and he had sex with me forcefully under my mother's watch and she was willing to beat me should I try to resist, it was so painful that was my first sexual encounter and I later found out that I am pregnant."*

*"I was made to sleep with the traditional healer and fell pregnant and each time I look at this pregnancy it makes me recall the whole abuse."*

### 3.2. Theme 2: Socioeconomic Factors–A Licence for Teenage Pregnancy and Freedom

Theme 2 had two subthemes, namely feelings of love and trust for the male counterpart and the need for security away from home, with the pressure for survival.

### 3.2.1. Subtheme 2.1: Feelings of Love and Trust for the Male Counterpart

A total of 8 out of the 20 participants indicated that feelings of love and trust for the male partner could lead to a teenager falling pregnant as a way of demonstrating love to the boyfriend. This factor was classified as socioeconomic because it combined economic and sociological measures of a teenager's access to resources on a financial basis and their standing in society.

*"Failing pregnant was proof enough to my boyfriend that I love him and thus condoms were not necessary, and he loves me too."*

*"True love should have trust, and this is what we had with my boyfriend, and I fell pregnant, unfortunately when reporting the incidence to my family they immediately escorted me to his parents."*

### 3.2.2. Subtheme 2.2: The Need for Financial Security

The adolescents explained that having sex with a rich man helped them to get money to take care of their basic needs, such as a place to stay when they were far away from home, more specifically, at tertiary institutions, when they were from a poverty-stricken family. This included having sexual relations with men who were old enough to be their fathers.

*"I did not have anywhere to go I had to have sex with this man in order to get a roof over my head."*

*"When I left home to the university my aunt only gave me money for the bus, she did not even bother where I was going to sleep and on arrival, I met this man who appeared to be very good to me and before I could think he wanted to have sex with me and I fell pregnant. And I was safe for a while staying with him in his room."*

*"I had a blesser who paid for my accommodation and gave me more money and I forgot about my poor background however he did not want us to use condoms and I felt pregnant."*

*"In order to have money to pay for my room at school I had to sleep with a man, and unfortunately this man emphasized that no condom at all, and I fell pregnant. And I was pregnant away from home I ended up leaving school coming home."*

Based on the socioeconomic status of the adolescent girls, certain cultural factors are relevant, as indicated in the next theme.

### 3.3. Theme 3: Cultural Factors to Which Adolescent Girls Are Susceptible

Theme 3 had two subthemes: menarche, the first, is a sign of maturity and readiness for marriage, and the second is the forced or arranged marriage to a rich man as a source of wealth for an adolescent girl and the family.

### 3.3.1. Subtheme 3.1: Menarche Is a Sign of Maturity and Readiness for Marriage

Most adolescent girls became emotional while talking about this subtheme. The participants indicated that menarche is culturally recognized as a sign of maturity, which signifies readiness for marriage. The participants highlighted that immediately after reporting their menarche to the family, preparations were made for them to go to female initiation school.

> *"One day I woke up with blood between my thighs and communicated with my family and friends, and I was told I am ready for marriage and at that time it didn't make sense to me. However later in the year, my parents sent me to female initiation, and I was formally told I am a woman I can now get married."*

### 3.3.2. Subtheme 3.2: Forced or Arranged Marriage to a Rich Man as a Source of Wealth for an Adolescent Girl and the Family

The participants indicated that the financial status of their families forced them to marry rich men in order to have access to the funds necessary for their family's upkeep. One participant was forced to get married to a rich man, and two others decided to do so of their own free will as their only source of income, and later, an arranged marriage without consultation with the girl was also made. The forced marriage was either because the family owed the man some money or the adolescent girl was used as collateral.

> *"I am only 16 and I was forced to get married to an old man to be his fourth wife because he is rich and he will support my family financially, and immediately after the marriage I fell pregnant and my parents were excited."*

> *"I am an orphan and I have to take care of my siblings so the best option was to get married to a rich man be pregnant for him then and then I and my siblings will be able to have a source of money to meet our needs."*

> *"The grant we are getting it's not enough so the other way of survival is to get married to a man who has more money, and this will help you support your family by buying them groceries every month end."*

> *"I am a teenage mother not by choice but pressure from my uncles to get married to a man they owed him money and as an orphan, I did not have any choice but to succumb and my brother was treated differently as a man there was no pressure for him to get married."*

> *"You won't believe what happened I was forced to get married to an old man and that's how I ended up with this child at my age."*

Getting married at an early age and falling pregnant as a teenager has consequences, as presented in the last theme.

### *3.4. Theme 4: Consequences of Teenage Pregnancy*

This theme had four subthemes, as presented below. Some of the teenagers shed tears while talking about their experiences as pregnant teenage girls.

### 3.4.1. Subtheme 4.1: How Can I Turn Back the Clock? Depression and Feelings of Regret

Adolescent girls showed signs of frustration when talking about the experiences of ending up pregnant, not knowing how to reverse the whole scenario and restart their lives.

> *"I was forced to marry a man I did not know nor love which made me battle with my feelings and ended up with depression, do you imagine having sex with someone you don't love, it's so painful."*

> *"I regret everything I don't know how best I can change it, being forced to have sex with a stranger for monetary benefits for my parents, I ended up pregnant and the baby will serve as a reminder for the rest of my life."*

### 3.4.2. Subtheme 4.2: Sex as a Mode of Payment; a Traumatic Experience

One adolescent stated that she was forced to have sex with the traditional healer at the age of 15 as a means of payment for the treatment offered to her late mother by this traditional healer. She fell pregnant by the traditional healer, and she indicated that she was traumatized by this experience.

> *"I was forced to have sex with the traditional healer who have treated my mother before her death as a mode of payment for the services when treating her and this resulted in me falling pregnant at the age of 15 a traumatic experience I will never forget for the rest of my life."*

### 3.4.3. Subtheme 4.3: Hatred towards Perpetrators

One participant indicated that she was locked in a room together with the stepfather in her own home and was raped with her mother's knowledge. She fell pregnant, and she felt hatred towards her mother and stepfather.

> *"My mother forced me to have sex with my stepfather after she locked me in the same room with him. I hate them (my mother and my stepfather)"* (tears pouring from her eyes).

## 4. Discussion of Findings

The discussions with the adolescent girl participants revealed that these girls experience serious challenges related to teenage pregnancy. However, different experiences are unique to adolescent girls.

### 4.1. Socioeconomic Factors—A Licence for Teenage Pregnancy and Freedom

The findings involve socioeconomic factors which make adolescent girls see teenage pregnancy as a license to economic freedom. Others see teenage pregnancy as social security to prove to their sexual partners that they love and trust them by falling pregnant.

These findings are in line with the study conducted by Neville [10], which found that teenagers become pregnant because they believe that pregnancy will stop their boyfriends from leaving them and that being a mother will give them a sense of fulfillment.

The teenagers surveyed in this current study explained that having sex with a man helped for safety and security reasons when they were far away from home, more specifically, at tertiary institutions and when they were from a poverty-stricken family. This included having sexual relations with men who were old enough to be their fathers. Okechukwu et al. [11] revealed that peer pressure, sexual abuse, and poor knowledge of sex are major contributing factors that lead to teenage pregnancy in secondary schools. Sex education should be promoted in families, schools, and traditional and religious organizations. According to a study conducted by Akella & Jordan [12], poverty is one of the issues leading to teenage pregnancy, which is in line with the findings of this study.

According to Chirwa et al. [13], teenage pregnancy is a driver of socioeconomic inequalities as it worsens poverty among rural disadvantaged communities. Their study revealed a high concentration of teenage pregnancy among poor households.

### 4.2. Cultural Factors to Which Female Teens Are Susceptible

The study's findings in this theme were simply divided into these two categories: menarche is a sign of maturity and readiness for marriage; getting married to a rich man is a source of wealth rather than education for a girl.

Ibitoye et al. [14] explained that although the links between early menarche and sexual and reproductive health in LMICs are likely to be similar to those found in HICs, there are differences in the sociocultural factors related to menarche and sexual and reproductive health [14]. For example, in many LMICs, menarche has traditionally served as a cultural rite of passage and marker of adulthood, positioning a girl as ready for marriage. Thus, early menarche is more likely to be associated with early marriage in LMICs. This is not the case in most HICs. Furthermore, in many LMICs, girls' mobility and social interactions are

often restricted once they reach puberty, limiting their opportunities to engage in premarital sexual activity.

The Limpopo provincial population and development directorate report [15] revealed that there are several factors that are associated with teenage pregnancy in Limpopo: e.g., exposure to sex and psychosocial, economic, cultural, and household factors. The three economic factors identified in this current study were poverty, at 61.3%, followed by child support grants (37.0%), and intergenerational relationships, at 11.8%. Many teenage girls get pregnant to escape poverty but then miss a lifetime opportunity of poverty emancipation through education. These challenges, coupled with teenage pregnancy, lead to the destruction of girls' lives, whereas a male teenager fathering the baby continues with his education. This finding is in line with a study conducted by Uwizeye et al. [16] in Rwanda, which found that the socioeconomic conditions of teenage girls can lead to teenagers being impregnated by men, specifically men with good financial resources.

The United Nations Secretary-General discouraged child marriage and classified it as "a violation of human rights. By 2020, 142 million innocent young girls worldwide will be separated from their friends and family, deprived of an education, and put in harm's way because of child marriage. Together, let us resolve to end the discrimination and poverty that perpetuate this harmful practice. And let us help those who are already married to lead more fulfilling lives. All members of society will benefit when we let girls be girls, not brides"—United Nations General Assembly [17].

The Executive Director of the United Nations Population Fund said that child marriage "is an appalling violation of human rights and robs girls of their education, health, and long-term prospects. A girl who is married as a child is one whose potential will not be fulfilled. Since many parents and communities also want the very best for their daughters, we must work together and end child marriage" [18]. In many societies, marriage is a celebrated institution signifying a union between two adults and the beginning of their future together. "Unfortunately, millions of girls still suffer from a vastly different marriage experience every year. Worldwide, many brides are still children, not even teenagers. So young are some girls that they hold onto their toys during the wedding ceremony. Usually, these girls become mothers in their early teens, while they are still children themselves" [18].

*4.3. Consequences of Teenage Pregnancy*

The findings of the study under this theme yielded the emergence of only one category, as presented below. Some of the teenagers surveyed shed tears while talking about their experiences related to their pregnancy.

A study conducted by Mangeli et al. [19] reveals that teenagers feel emotional and mental distress and are disappointed by teenage pregnancy and the transition into motherhood [19]. The findings of this study support this: some adolescents had feelings of regret because they were not ready to take care of a baby. One adolescent felt helpless and despair due to the failure to adjust to the implications of her pregnancy. These findings are similar to those of a study conducted by Akella & Jordan [12], in which they revealed that teenage pregnancy could result in feelings of despair and hardships, such as depression, which was experienced by one of the participants in this study [12].

Wall-Wieler et al. [20] indicate that teenage pregnancy has several consequences, e.g., health consequences coupled with high risks of maternal death, high risk of obstetrics complications, low birth weight, and a high risk of infant mortality. Educational consequences include dropping out of school, absenteeism, and poor academic performance. This is further supported by a study conducted by Ramalepa et al. [21] in primary and secondary schools in South Africa, where teachers were concerned about the educational success of pregnant learners, the disruption to learning, school dropout, absenteeism, and a lack of school-based health services to provide prevention and support services to teenagers [21]. Economic consequences include lower family income, leading to an increased dependency ratio, thus aggravating poverty. The social consequences are stigma, discrimination, less likelihood of getting married, and a strong possibility of suffering abuse [22].

Pregnant teenagers face challenges related to pregnancy that are similar to those faced by other women; however, they tend to have additional concerns. Specifically, those under 15 years suffer biological issues of not being developed enough to sustain a healthy pregnancy to give birth. Some complications experienced include low birth weight, premature labor, anemia, and pre-eclampsia, which are connected to biological age and are observed in teen births even when other risk factors are under control [23,24]. Teenage pregnancy also predisposes them to the risk of contracting HIV and the transmission of HIV from mother to baby, which is a major concern for South African women's health [25].

Ajayi & Ezegbe [26] revealed that teenagers from communities where there is sexual violence are more likely to also experience gender-based and sexual violence, have an unintended pregnancy, and become vulnerable to HIV [26]. Gender-based violence or the fear of violence can stop women and girls from negotiating safer sex, accessing HIV and sexual and reproductive health services, and disclosing their HIV status to partners, family members, and health providers. Artz et al. [27] also found that 12% of women reported forced sex without condoms, sexual violence, or rape as a reason for seeking HIV testing; these acts were perpetrated mostly by their intimate partners [27]. Women and girls who are survivors of violence suffer a range of health consequences, including mental health issues, such as depression and anxiety, higher use of alcohol, less control over sexual decision-making, and poor sexual and reproductive health outcomes. Studies show that women living with HIV who have experienced intimate partner violence are significantly less likely to start or adhere to antiretroviral therapy and have worse clinical outcomes than other women living with HIV [28]. Women and girls who experience violence are also less likely to adhere to both pre-exposure and postexposure prophylaxis [29,30]. In some regions, women and girls who have suffered intimate partner violence are 1.5 times more likely to acquire HIV than women who have not suffered this type of violence [31]. More than one in three women and girls worldwide have experienced physical and/or sexual violence, often at the hands of their intimate partners [32]. A global review found that women who have experienced violence are 16% more likely to have a baby with low birth weight and almost twice as likely to experience depression [1,31].

The rates of teenage pregnancies are higher in societies where it is traditional for girls to get married when they are young and where they are encouraged to bear children as soon as possible. In some sub-Saharan African countries, early pregnancy is often seen as a blessing because it is proof of the young woman's fertility [33]. Countries where teenage marriages are common experience higher levels of teenage pregnancies. On the Indian subcontinent, early marriage and pregnancy are more common in traditional rural communities than in cities.

Multiple studies have indicated a strong link between early childhood sexual abuse and subsequent teenage pregnancy in industrialized countries related to human trafficking, exploitation, and all forms of violence. Up to 70% of women who gave birth in their teens were molested as young girls; in contrast, 25% of women who did not give birth as teens were molested [1,25]. The expectations of communities of girls to become mothers at a young age can be a cause of teenage pregnancies, along with sexual violence. Other factors to be considered include rape and early marriages of children to older men, coupled with unique risks faced by these girls during pregnancy, such as interruption to education and health risks, such as HIV, premature birth, and increased maternal mortality, denying the girls the right to live a healthy life. All these issues are related to gender inequality [34].

*4.4. Recommendations*

A holistic approach is required to deal with teenage pregnancy. This means not focusing on changing the behavior of girls but dealing with the underlying reasons for adolescent pregnancy, such as poverty, gender inequality, social pressures, and coercion. This approach should include providing age-appropriate comprehensive sexuality education for all young people, investing in girls' education, preventing child marriage, sexual violence, and coercion, building gender-equitable societies by empowering girls and engaging men

and boys, and ensuring adolescents' access to sexual and reproductive health information, as well as services that welcome them and facilitate their choices.

Community education may also help to provide knowledge of the sexual reproductive rights of a child girl. In communities where early marriages are arranged for teenagers, there might be a lack of knowledge or abuse of cultural norms and beliefs. Comprehensive responsible sexual behavior education and improved dual contraceptive counseling, delivery, and access, including emergency contraception or morning-after pills and female and male condoms, are also recommended. The use of Implanon and IUCDs might be suitable for women who experience gender-based or intimate partner violence. Dialogue on and the awareness of contraceptives and biomedical prevention methods for HIV, such as contraceptives, PREP, and PEP, are now available for free in the public sector.

The capacity building of the primary healthcare re-engineering team must empower community health workers to provide counseling and education on the prevention of teenage pregnancy during their household support visits to the entire family.

*4.5. Contribution/Implications of the Study*

The study findings can be used by policymakers to review children and adolescent legislation, policies, and programs to improve the life of these age groups and contribute meaningfully to the socioeconomic and cultural aspects of the community. The study might improve the life skills, self-esteem, and confidence of teenagers in dealing with teenage pregnancy and sexual reproductive health in general.

## 5. Limitations

The study was limited to Mopani District Municipality and cannot be generalized to other municipalities. Therefore, further studies could be conducted in other municipalities in the future to have a better understanding of what is happening in the whole district. The study used one study approach, i.e., qualitative, but the use of a quantitative approach might also yield more study results.

## 6. Conclusions

The study highlights the serious challenges experienced by girls regarding teenage pregnancy. Socioeconomic factors are highlighted as one of the critical factors when it comes to teenagers regarding the extent to which they opt to engage in sexual activities with older men for financial gains. The cultural factors responsible for fueling teenage pregnancies in communities should be addressed. However, some of these factors are interconnected with family background, such as poverty, and hence, teenage pregnancy keeps on escalating. Unfortunately, teenage pregnancy has its own consequences which last for a lifetime, such as having a baby at an early age, serving as a long-term reminder regarding those experiences. These challenges must be dealt with to improve the lives of women in the Mopani District in the future. If they are neglected, the impact will be felt too late, and the complications will be irreversible. Prevention is better than cure.

**Author Contributions:** Conceptualization, S.H.M. and methodology, P.R.R.; formal analysis, P.R.R.; investigation, P.R.R.; data curation, P.R.R. and S.H.M.; writing—original draft preparation, P.R.R. and S.H.M.; writing—review and editing. All authors have read and agreed to the published version of the manuscript.

**Funding:** This research received no external funding.

**Institutional Review Board Statement:** The study was conducted in accordance with the Declaration of Helsinki, and approved by the Institutional Review Board (or Ethics Committee) of College of Human Sciences research ethics committee (protocol code 90187598_CREC_CHS_2021).

**Informed Consent Statement:** Informed consent was obtained from all subjects involved in the study.

**Data Availability Statement:** The data presented in this study are available on request from the corresponding author.

**Conflicts of Interest:** The authors declare no conflict of interest.

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
