# Peer review of "Adolescent Girls’ Experiences Regarding Teenage Pregnancy in the Rural Villages of Limpopo Province, South Africa"

_adolescents, doi:10.3390/adolescents3010004_

Round 1

Reviewer 1 Report (Previous Reviewer 2)

The suggestions were accepted. In my opinion, the modifications made allowed the manuscript to be more consistent and ready to be published.

Author Response

Dear reviewers

Thanks for the review, attached find the review report

Kind regards

Prof Risenga

This manuscript is a resubmission of an earlier submission. The following is a list of the peer review reports and author responses from that submission.

Round 1

Reviewer 1 Report

Abstract:

In the abstract unbold the word Every that starts the abstract and add a space between Abstract: and Every.

Introduction: 

Do not start a sentence with numerical values, they should be written as words. In the South Africa paragraph, the girls mentioned of 10 years of age or younger are adolescents not teenagers and this ought to be corrected.

The major concern is the literature review is too small and it would be expected for such a topic more theory and background information increasing the references up to 50-60 or more would be necessary to create a more informed approach to this topic.

Section 2.2., remove the bullet points and write out as full sentences.

Section 2.4 for full sample size it should be N = 20 and for subsamples should use small n

Section 4.1 and 4.2 the beginnings of each sentence need to be re-written as starting with the reference in-text citation as a bracket is incorrect format.

The study limitations and conclusions should be expanded to help generalize or contrast what had been learned from the present study and what can be used to inform other countries of similar risk factors that young teenage girls might face and what can be proactively done to help reduce those risks as a universal recommendation.

Reviewer 2 Report

Comments are in the attached file.
